# Understanding the Spatiotemporal Characteristics of Land Subsidence and Rebound in the Lianjiang Plain Using Time-Series InSAR with Dual-Track Sentinel-1 Data

Yangfan He [1], Alex Hay-Man Ng [1,2,*], Hua Wang [3] and Jianming Kuang [4]

1. Department of Surveying Engineering, Guangdong University of Technology, Guangzhou 510006, China; 2112109043@mail2.gdut.edu.cn
2. Key Laboratory for City Cluster Environmental Safety and Green Development of the Ministry of Education, Guangdong University of Technology, Guangzhou 510006, China
3. College of Natural Resources and Environment, South China Agricultural University, Guangzhou 510642, China; ehwang@gdut.edu.cn
4. School of Civil and Environmental Engineering, the University of New South Wales (UNSW), Sydney, NSW 2052, Australia; jianming.kuang@student.unsw.edu.au
* Correspondence: hayman.ng@gmail.com or hayman.ng@gdut.edu.cn

**Abstract:** The Lianjiang Plain, renowned for its position as 'China's textile hub' and characterized by its high population density, has experienced considerable subsidence due to excessive groundwater extraction in recent years. Although some studies have investigated short-term subsidence in this plain, research on long-term subsidence and rebound remain understudied. In this paper, the characteristics of surface deformation in the Lijiang Plain during two periods (2015–2017 and 2018–2021) have been investigated using the time-series interferometric synthetic aperture radar (TS-InSAR) technique, and the correlation with the changes in groundwater level, geological factors, and urban construction are discussed. The InSAR-derived results are cross-validated with the adjacent orbit datasets. Large-scale and uneven subsidence ranging from −124 mm/year to +40 mm/year is observed from 2015 to 2017. However, a significant decrease in the subsidence rate during 2018–2021, with local rebound deformation up to +48 mm/year in three regions, is also observed. Groundwater level changes are found to be the major cause of the ground deformation, and the intercomparison between groundwater level and ground displacement time series from TS-InSAR measurements also indicates a clear relationship between them during 2018–2021. Geological factors control the range of deformation area over the study period. The impact of urban construction on surface subsidence is evident, contributing to high deformation. Our findings could improve the understanding of how deformation is affected by groundwater rebound and offer valuable insights into groundwater management, urban planning, and land subsidence mitigation.

**Keywords:** land displacement; time-series InSAR; groundwater level; Lianjiang Plain

## 1. Introduction

Land subsidence with natural or anthropogenic origin is a global phenomenon and may cause environmental and socioeconomic impacts [1–4]. The Lianjiang Plain (LJP) is one of the important manufacturing centers for knitted underwear in China [5]. Relying on the strengths of its knitting industry, the economic and urban development of this plain have been rapidly increasing over the last two decades, with gross domestic product (GDP) rising from CNY 25.9 billion in 2000 to over CNY 151.5 billion by 2020, an average annual increase of 0.9%. At the same time, printing and dyeing activities pollute surface water, reduce the availability of clean water, and put increasing pressure on limited groundwater resources and the ecological environment [6]. In addition, it has been affected by land subsidence but is overlooked or underestimated because of its slow developmental characteristic. The exponential population increase and the further development of industry in the Plain have

led to a significant increase in water demand. Subsidence continued as a direct result of groundwater extraction far exceeding the natural recharge [7]. Long-term subsidence not only has led to the destruction of houses and the formation of earth fissures but also makes it more vulnerable to flooding during rainy seasons. Over the past eight years, the LJP has frequently encountered extreme surface flooding when intensive rainfall took place, affecting houses, basements, and other underground structures [8–10]. Frequent flooding and continued land subsidence in the LJP killed 12 people, caused the collapse of 20 thousand hectares of crops, produced direct economic losses reaching up to CNY 3.5 billion, and put the lives of one million people at risk [11,12]. Therefore, systematically monitoring and analyzing the long-term spatiotemporal evolution of land deformation is vital for disaster prevention and warning as well as promoting viable and sustainable development.

To date, few deformation studies have been conducted in the LJP using point-based surveying methods, including leveling surveys and Global Positioning System (GPS), because of their high cost and time-consuming nature [13]. Therefore, there is a need for accessible and effective methods to monitor the distribution and status of land deformation in the LJP. Interferometric Synthetic Aperture Radar (InSAR), an ideal and effective remote sensing method for monitoring ground deformation, has been widely utilized for this purpose. InSAR has already been successfully applied for monitoring a range of ground displacement applications over the past 20 years, including glacier motion [14], earthquake deformation [15,16], volcanic movement [17], fluid flow [18,19], slope stability monitoring [20], mining activities [21], etc. Compared to conventional geodetic monitoring techniques, InSAR offers a low-cost and low-labor way to measure surface displacement over large spatial coverages with a high accuracy of centimeters or even millimeters [22–24]. TS-InSAR, an extension and supplement to InSAR, has been developed to map time-series land surface displacement, including Persistent Scatterer Interferometry (PSI) [22] and Small Baseline Subset (SBAS) [25]. The principle of TS-InSAR is to improve deformation measurement accuracy through using multiple SAR images and significantly reducing the drawbacks in conventional differential InSAR (D-InSAR) [22,26,27]. In addition to the TS-InSAR methods mentioned above, many other approaches have been developed for different monitoring conditions, which can be divided into two main categories: PS-based methods that work on single point targets [3,28,29], and small baseline-based methods that utilize spatially distributed targets [25,30,31]. All of these TS-InSAR methods have facilitated ground subsidence monitoring in both urban and rural areas using various SAR datasets [3,29,32–36].

Past studies of the LJP, based on TS-InSAR methods using various SAR data, have found regional scale subsidence between 2006–2011 and 2015–2018. These studies have demonstrated the feasibility of the TS-InSAR method for determining surface displacement in the LJP. In addition, these studies have revealed the influence of structural geology, land use, and faults on land subsidence [13,37]. Subsequent land subsidence was also observed by Zhang et al. [38]. In particular, two obvious land uplift zones in the northern part of the plain were detected. Zhang et al., through their analysis of displacement results with geological data and local government reports, arrived at the conclusion that groundwater was the major factor causing combined land subsidence and rebound. More recently, Huang et al. [39] examined ground deformation and evaluated the deformation induced by human activities using interannual groundwater extraction data from 2015 to 2020. However, the above-mentioned studies detected and analyzed subsidence in a specific time from 2015 to 2021. Most of these researchers emphasized the linkage between annual groundwater extraction and land subsidence. However, the spatiotemporal evolution of subsidence and rebound, especially in response to local-scale groundwater levels, were not thoroughly investigated. The influence of groundwater rise on regional subsidence and rebound deformation at this site remains inadequately understood. Moreover, there is a clear need for further refinement in the quantitative analysis of deformation characteristics caused by natural and anthropogenic factors.

This paper is built upon the previous literature to conduct a comprehensive deformation study over the LJP to obtain better understanding of the deformation phenomenon at the site. The main objectives of this study are as follows: (a) to quantify and map the latest deformation trends of the LJP, utilizing long-term time-series deformation data derived using TS-InSAR (2015–2021); (b) to investigate the relationship between deformation and multiple influencing factors through collecting in situ hydrogeological data and multi-temporal building data, with a particular focus on the quantitative relationships between groundwater levels, geological factors, and urban construction. The paper is organized as follows. First, the long-term deformation velocity in the LOS (Line-Of-Sight) direction during 2015–2017 and 2018–2021 is calculated by means of applying the TS-InSAR analysis to unveil the spatio-temporal evolution of land subsidence and rebound in the LJP. Second, cross-validation of the overlap areas between adjacent orbits is conducted. Finally, subsidence-influencing factors such as groundwater levels, geological conditions, and urban construction are taken into account, and their correlation with deformation is thoroughly analyzed.

## 2. Study Area and Datasets

### 2.1. Study Area

The Lianjiang Plain is located in the southern part of the Chaoshan Alluvial Plain in Guangdong Province (Figure 1). The area of the Plain is more than 3000 square kilometers, which spans across three administrative regions: Chaonan (687 km$^2$), Chaoyang (843 km$^2$), and Puning (1615 km$^2$). The Plain is characterized by a subtropical monsoon climate and is well-endowed with abundant sunshine and rainfall. According to local governmental agencies' records, the annual rainfall is 1750 mm. The spatial and temporal distribution of rainfall in the Plain is non-uniform, with more rainfall observed in the southeast than in the northwest, and more frequent rainfall between June and September.

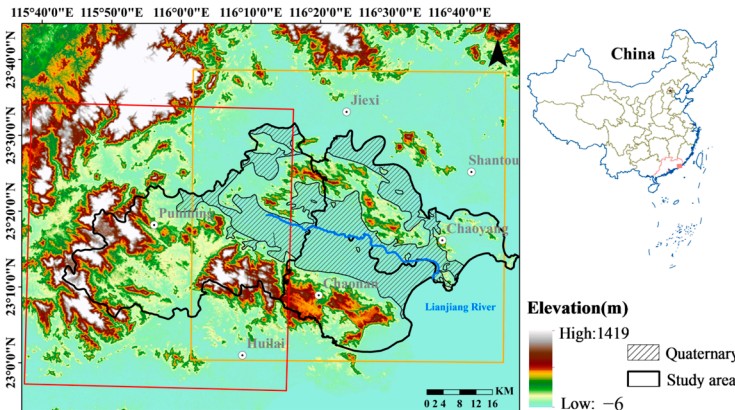

**Figure 1.** Geological formation of the Lianjiang Plain and data coverage of the SAR data. The yellow and red rectangles are the Sentinel-1A tracks 40 and 113 coverage, respectively. The light pink box in the top-right-corner map shows the location of the study area in China.

The Lianjiang River basin flows through the Lianjiang Plain and serves as the main river in the area. Since the 1990s, the Lianjiang River has been subjected to severe pollution resulting from the discharge of waste by printing and dyeing companies and electronic waste [6]. Approximately 5.34 million inhabitants live within this plain, with an average population density of approximately 1700 inhabitants/km$^2$. The area is severely challenged by scarce water resources, with surface water resources per capita only one-fifth of those in Guangdong Province. The LJP is highly dependent on groundwater resources, with extensive extraction of groundwater reported in many years [40]. Since 2018, the provincial government has been leading efforts towards comprehensive pollution control management in the Lianjiang River, resulting in a significant improvement in water quality [41]. Meanwhile, water conservation policies and measures aimed at restricting water consump-

tion in printing and dyeing firms have also played a role in significantly reducing the amount of groundwater extraction.

The Plain is characterized by flat terrain, and the elevation of the Plain ranges from 0 to 967 m. Extensive deposition processes occurred in the Lianjiang Plain, with geological studies indicating that a larger part of the Plain is covered by Quaternary alluvial sediments, while the remaining area is overlaid by bedrock, as presented in [42]. The thickness of the Quaternary deposit, determined via interpolation of detailed borehole data [42], genereally ranges from 40 m to 130 m. The locations of boreholes are represented with light blue triangles in Figure 2.

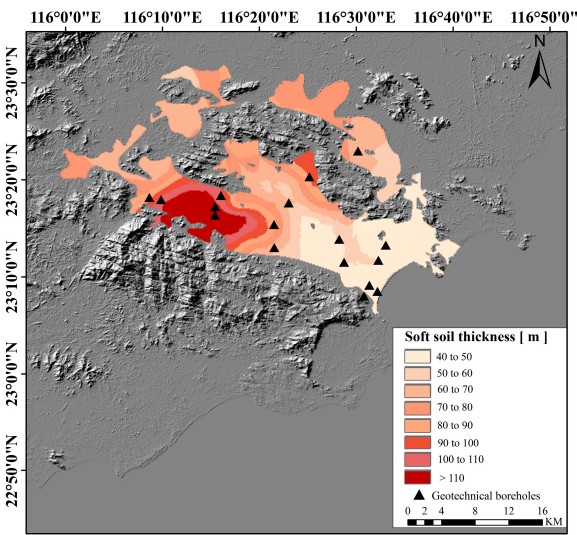

**Figure 2.** Sedimentary thickness map and borehole locations. The thickness of the sediment has been calculated by means of geotechnical borehole data.

### 2.2. Data

A total of 129 Sentinel-1A SAR images along two adjacent orbits, covering the period from June 2015 to November 2021, were collected and used. The coverage of SAR images is depicted in Figure 1. Specific imaging parameters and properties of Sentinel-1A datasets are illustrated in Table 1. Detailed distribution information regarding the spatiotemporal baselines of these datasets is presented in Figure 3. The master image used for datasets I–III was acquired on 6 May 2016, 3 February 2020, and 8 February 2020, respectively.

**Table 1.** Specific parameters of Sentinel-1 datasets.

| Datasets | Dataset I | Dataset II | Dataset III |
|---|---|---|---|
| Incidence angle | 34.14° | 37.06° | 39.48° |
| Track | 40 | 40 | 113 |
| Orbit direction | Ascending | Ascending | Ascending |
| Polarization | VV | VV | VV |
| Number of Scenes | 33 | 48 | 48 |
| Time range | 2015–2017 | 2018–2021 | 2018–2021 |

Furthermore, geological boreholes were utilized to determinate the depth of Quaternary deposits in the Plain. Groundwater level data from six water wells [43], with a sampling period of six times a month, were used to assess the relationship between InSAR-derived surface deformation and groundwater level measurements.

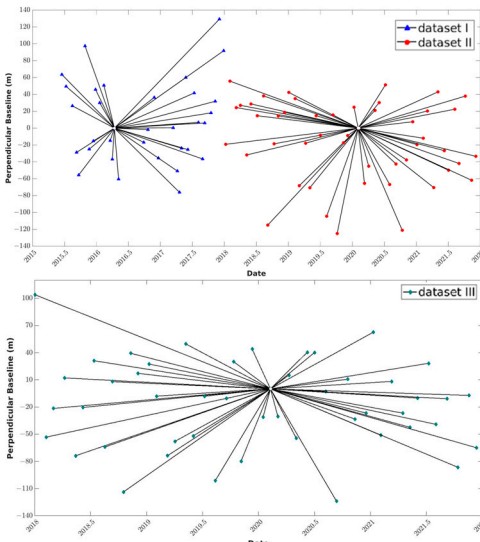

**Figure 3.** Baseline information for the Sentinel-1 datasets.

## 3. Methodology

### 3.1. InSAR Time Series Processing

In this study, deformation information was derived from the multi-temporal Sentinel SAR data, utilizing the PSI approach based on InSAR time-series analysis software named GEOS-ATSA (Advance Time-Series Analysis) [28,44]. The interferograms were first generated using the conventional DInSAR method based on the InSAR Scientific Computing Environment (ISCE) software [45], where the topographic phase component was estimated and removed using the 1-arc second SRTM DEM [46]. After generating the interferograms, initial PS candidate point (PSC) selection based on amplitude analysis was performed. Pixels with dispersion amplitude index (DA) lower than 0.25 were considered as reliable PSCs and are included in the following analysis. A triangular irregular network (TIN) was then established via connecting the selected PSCs with a maximum arc length of 1.5 km. The phase difference between two pixels within each arc of the TIN can be written as:

$$
\begin{aligned}
\Delta\varphi_{diff} &= \Delta\varphi_{defo} + \Delta\varphi_{topo} + \Delta\varphi_{APS} + \Delta\varphi_{Noise} + \Delta\varphi_{orbit} \\
&= -\frac{4\pi}{\lambda}B_t\Delta v - \frac{4\pi}{\lambda}\frac{B_\perp}{rsin\theta}\Delta h + \Delta\varphi_{APS} + \Delta\varphi_{Noise} + \Delta\varphi_{orbit}
\end{aligned}
\tag{1}
$$

where $\Delta\varphi_{defo}$ refers to the phase of displacement, $\Delta\varphi_{topo}$ denotes the topography phase, $\Delta\varphi_{APS}$ indicates the phase due to atmospheric disturbances, $\Delta\varphi_{Noise}$ is the noise phase, $\Delta\varphi_{orbit}$ represents the phase due to residual orbital error, $\Delta v$ and $\Delta h$ denote the difference of displacement rate and topographic error, and $B_t$, $B_\perp$, $r$, $\lambda$, and $\theta$ are the temporal and perpendicular baseline, range distance, radar wavelength, and local incidence, respectively. In order to obtain the $\Delta\varphi_{defo}$ and $\Delta\varphi_{topo}$ phase components, which are the main objectives of InSAR processing, it is necessary to estimate and eliminate the remaining phase components. The estimation of the modeled parameters ($\Delta v$ and $\Delta h$) is hence carried out using the Least-squares Ambiguity Decorrelation Adjustment (LAMBDA) method [47]. Then, the obtained relative modeled parameters were converted to absolute values through performing the spatial integration and integration testing process. In addition, the less reliable PCSs (DA less than 0.4) were also included and analyzed in order to increase the number of PS. An adaptive estimation strategy was applied to estimate the parameters of these PCSs [48]. The sparse phase unwrapping method [49] was utilized to unwrap the residual phases. The linear spatial trend in unwrapped residual phases was assumed to be caused by the orbit errors and hence was removed. The remaining residual phase components include atmospheric artifacts, noise term, and non-linear displacement. The filtering operations were executed in spatial–temporal mode to separate the nonlinear

deformation phase from the two components. At the end, the time-series deformation results were derived. The data process flowchart is shown in Figure 4.

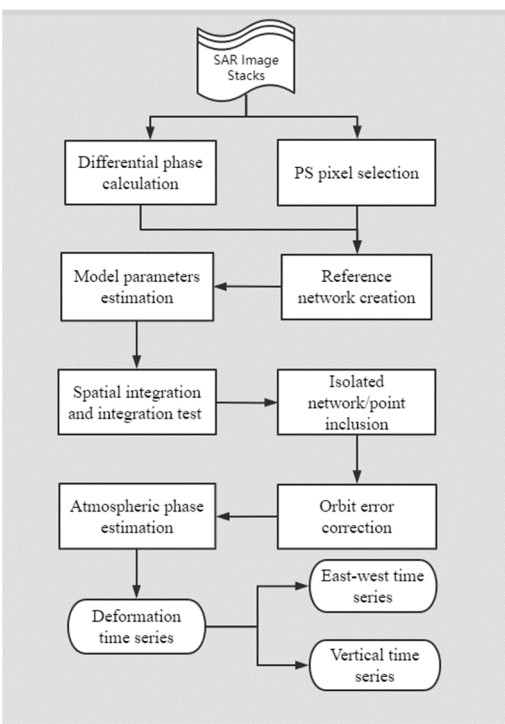

**Figure 4.** Data process flowchart.

### 3.2. Estimated Vertical and Horizontal Deformation

According to the aforementioned InSAR processing method, the resulting estimated displacement is in the LOS direction of the satellite. In areas where spatiotemporal coverage overlaps, multi-track acquisition geometries allow us to reconstruct real deformation in three dimensions: up–down, east–west, and north–south components. As a result, we calculated the displacement rates in the vertical (up–down) and horizontal (east–west) deformation fields, assuming that the component of the north–south is negligible [50,51]. Displacement components are computed using the following expression [52]:

$$D_{los} = \begin{bmatrix} cos\theta_{inc} & sin\theta_{inc} \end{bmatrix} \begin{bmatrix} D_u \\ D_e \end{bmatrix} \tag{2}$$

where $D_{los}$ is the deformation in the LOS direction and $D_u$ and $D_e$ are the deformation in the up and east directions. $\theta_{inc}$ is the radar incidence angle.

## 4. Results

### 4.1. Spatial-Temporal Variation of Surface Deformation

The annual LOS deformation rates of the study area for two periods, 2015–2017 and 2018–2021, were obtained through the TS-InSAR analysis. All results are adjusted based on a reference point, which was set in a relatively stable area as depicted by the red pentagram in Figure 5. As per convention, positive values in blue were deemed as the uplift, while negative values in red represented subsidence.

Figure 5a,b show the annual deformation rate for the periods 2015–2017 and 2018–2021, respectively. The comparison between the two results suggests that the study area is experiencing inconsistent surface deformation patterns. As show in Figure 5a, large-scale subsidence phenomena (T1), with a maximum rate of −124 mm/year, were recognized, affecting the central part of the Plain where proximity to faults, land use, and thickness of the Quaternary sedimentary were identified as the drivers. Meanwhile, the result of

2018–2021 shows a deceleration in the rate of subsidence, and in some areas, significant land uplift behavior is observed. Three main uplift zones were detected in the northwest and northern parts of the study area, marked as S1 (Puning), S2 (Chaonan), and S3 (Chaoyang) in Figure 5b, which had experienced subsidence during 2015–2017. The uplift rate ranged from +10 mm/year to +50 mm/year, with S1 having higher uplift values (+48.9 mm/year) than the other two areas, and involving a significant spatial magnitude of the uplift pattern. These broad uplifting zones are expected to be closely related to the recovery of groundwater, which will be discussed in detail in Section 5.2. In addition, the ongoing subsidence zone is persisting in the north-central part of the study area, with the maximum subsidence rate of approximately −45 mm/year, reducing by approximately 64% with respect to the period of 2015–2017. This is most likely be the consequence of a reduction in groundwater withdrawal due to government policy for effective groundwater management and active land subsidence prevention. This finding aligns with the result that groundwater pumping volume in this area has declined from more than 39 million cubic meters in 2015 to less than 22 million cubic meters in 2021 [52], as shown in Figure 6.

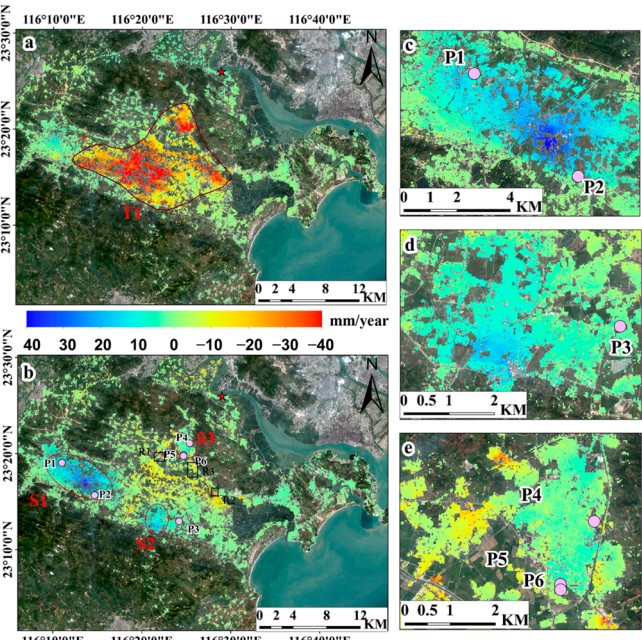

**Figure 5.** InSAR average deformation rate maps in LOS direction in LJP superimposed on Sentinel-2 image: (**a**) for the 2015–2017 period and (**b**) for the 2018–2021 period. The red circles (T1, S1–S3) delineate the extent of the deformation regions. The light purple dots (P1–P6) represent the locations of observation wells. The black rectangles (R1–R3) represent the land subsidence induced by urban construction. The enlarged S1, S2, and S3 uplift zones correlate to images in panels (**c**–**e**), respectively.

### 4.2. Cross-Validation

While the authors did not have access to the ground truth data over the site, the overlap time and coverage of dataset II and dataset III enabled validation of the InSAR results through comparing the common part of deformation maps from both datasets. The vertical deformation velocity derived from dataset III and dataset II, projected into the vertical direction assuming horizontal displacement is negligible, is shown in Figure 7a,b, respectively. The deformation results of the two maps are generally consistent in terms of deformation range and patterns. The velocity histogram for the two datasets is shown in Figure 7c; the correlation coefficient is 0.94, and the standard deviation of rates difference is 2.4 mm/year, which confirms the broad agreement between the two results. Furthermore, using the approach mentioned in Section 3.2, the vertical and horizontal deformation patterns during 2018–2021 are illustrated in Figure 8. The vertical velocity showed a similarity to LOS rates (Figure 5b). A slight east–west deformation was also observed. The

vertical component of the surface displacement dominates, suggesting that groundwater level is likely to be the trigger factor of land deformation over these areas. In Section 5.2, the vertical deformation time series within the overlapped regions will be used to explore their relationship with groundwater level changes.

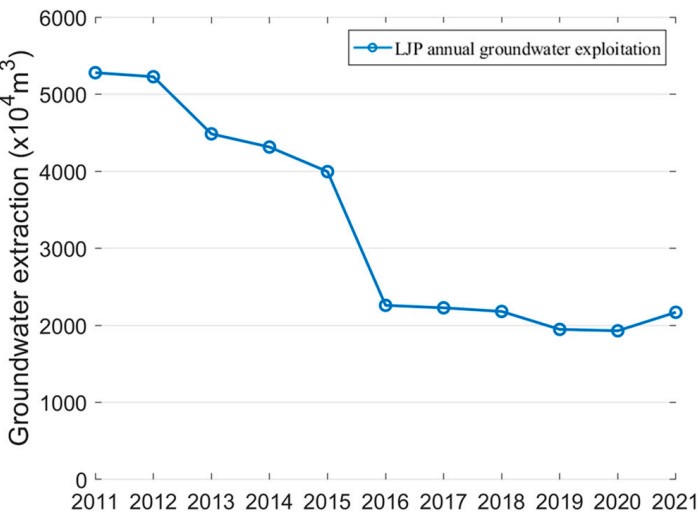

**Figure 6.** Annual groundwater exploitation in Lianjiang Plain.

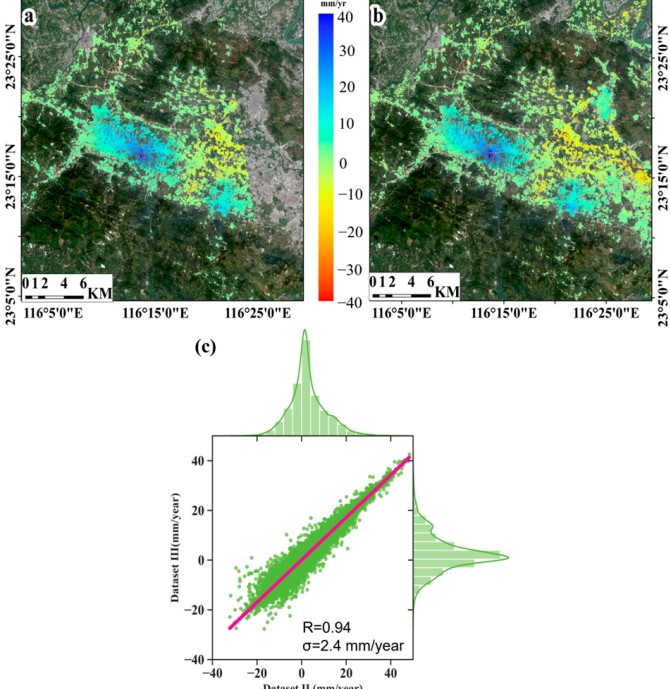

**Figure 7.** Average deformation velocity maps in the vertical direction acquired from dataset III (**a**) and dataset II (**b**) during 2018–2021. Cross-validation between deformation rates obtained from dataset II and dataset III (**c**). The red line is the fitting line of the linear regression. The upper and right green histograms denote the velocity distribution of dataset II and dataset III, respectively.

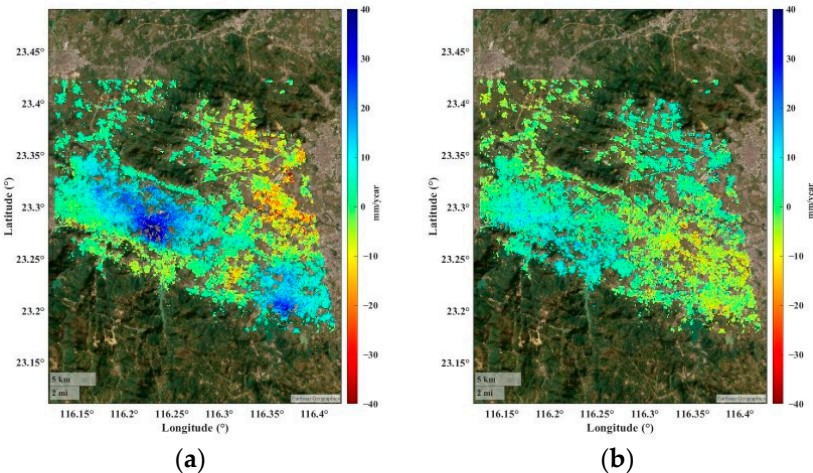

**Figure 8.** (**a**) Vertical rate and (**b**) horizontal rate from the combined datasets II and III.

## 5. Discussion

### 5.1. Comparison of Subsidence and Rebound between 2006–2021

Table 2 provides a comparison of this work and existing studies in terms of datasets, time span, and employed methodology. In general, the consistency in both spatial pattern and magnitude of the subsidence and uplift between this work and previous ones can be found. A regional subsidence bowl (T1) of centimeter order has been found at all sites in previous studies [13,37,53]. The two large uplift areas at Puning (S1) and Chaonan (S2) also coincide with previously detected uplifting bowls [39,40]. However, there are subtle differences when it comes to the magnitude of the deformation rate. The maximum subsidence detected in this study was located in Chaonan (T1), which is similar to the findings of Du et al. [37] and Liu et al. [13]. However, the peak subsidence estimated from this work (−124.4 mm/year) was slightly lower than in those studies (157.2 mm/year and −140 mm/year). Additionally, the maximum uplift was found in Puning (S1), occurring in agreement with the location of previous studies [38,39], while the peak uplift rate of +48 mm/year in this work differs from those, in which the uplift rate was over +100 mm/year and +20 mm/year, respectively. There are two possible reasons for the discrepancy in the observed peak subsidence and uplift rate: (1) different methods were employed to detect measurement points and estimate annual rates, such that more measurement points were detected in Liu et al. [13]; Du et al. [37] have greater subsidence due to longer wavelength compared to the band used in this study; (2) different observation periods contributed to different annual rates, and annual rates in Zhang et al. [38] are hence much higher than the average annual rates according to Huang et al. [39] and this study. In additional, Zhang et al. [38] and Huang et al. [40] noted the trend of stability for Chaoyang (S3), once a severe subsidence area, which is not consistent with the result of this study. The magnitude of the deformation Chaoyang (S3) varies from period to period, which could indicate temporal variation in deformation rates.

**Table 2.** Comparison of previous studies and this study.

| Reference | SAR Data | Location | Processing Method | Time Span |
|---|---|---|---|---|
| Du et al. [38] | ALOS | Guangdong Province | MT-InSAR | December 2006–October 2011 |
| Li et al. [54] | ALOS | Puning | SBAS-InSAR | December 2007–July 2010 |
| Liu et al. [14] | Sentinel-1 | Lianjiang Plain | DS-InSAR | November 2015– December 2017 |
| Zhang et al. [39] | RADASAT-2 Sentinel-1 | Lianjiang Plain | IPTA-InSAR | November 2018–December 2019 June 2015–December 2019 |
| Huang et al. [40] | Sentinel-1 | Chaoshan Plain | MT-InSAR | June 2015–October 2020 |
| This study | Sentinel-1 | Lianjiang Plain | GEOS-PSI | June 2015–December 2021 |

*5.2. InSAR-Derived Deformation Association with Groundwater Level Change*

The effective stress in the stratum can be altered due to changes in groundwater levels within underground aquifers, which may cause ground displacements at the Earth's surface [54,55]. Previous studies indicate that ground subsidence occurring in the LJP is most likely caused by extensive groundwater extraction [56]. Since 2018, the local government has implemented effective measures for groundwater conservation, leading to a decline in groundwater extraction and a possible recovery of regional groundwater to some extent. During groundwater recovery, subsidence rates are expected to slow down, and ground uplift is likely to be triggered once the recharged sections of the aquifer have regained the pore pressure in the stratum during the previous drawdown [57–59].

Information about annual groundwater extraction at the regional scale from 2015 to 2021 is represented in Figure 6. However, the regional scale data are unable to accurately portray the spatiotemporal evolution of groundwater because of their coarse resolution. To conduct a more thorough analysis of the spatiotemporal relationship between groundwater level and InSAR-derived deformation, groundwater level data from six local aquifer wells were gathered for the period 2018–2021. These data were subsequently used to form a time series. As such, groundwater level changes at the six groundwater observation wells (highlighted with purple dots in Figure 5) situated near the uplift zones were compared with the average ground deformation time series. The deformation time series was derived from all PS pixels within a radius of 150 m centered around each observation well. Given that the deformation data from multiple tracks are available for wells P1, P2, and P3, the vertical displacement time series obtained using the approach outlined in Section 3.2 is applied to these wells. For the rest of the wells, displacement data from the LOS direction are directly used for the comparison. The four-year groundwater level time series from the observation wells was plotted alongside the InSAR-derived deformation time-series measurements, as shown in Figure 9. A strong correlation was observed. Over a period of more than four years, a clear upward trend in groundwater level was observed for all wells, except P2, where no visible correlation was observed. Correlation coefficients were calculated, revealing the highest value for P1 (0.85) and P5 (0.79), followed by P3 (0.69) and P4 (0.61), while P6 displayed the lowest correlation (−0.44). It is worth noting that these observation wells are located at some distance from the center of the deformation zones. Consequently, the analysis was conducted using the groundwater level trends from observation wells near the uplift zone to understand the overall groundwater level trends in the uplift zone.

Within the S1 uplifting bowl, P1 and P2 are two representative points located inside of S1. The magnitude of cumulative vertical displacements measured at the P1 well and P2 well is 68 mm and 13 mm, respectively, during January 2018 and December 2021, with groundwater level change reaching up to approximately +7.95 m and −1.134 m for the two wells, respectively. Groundwater level evolution is aligned with the acceleration and deceleration of uplift at P1, but it is not the case for P2. There is a discrepancy between the InSAR results and the tendency of measured water level. There are three possible reasons: (1) the groundwater level monitoring well is too sparse in the area where coherence is already poor; (2) the magnitude of groundwater level change for P2 is not significant enough; (3) the two wells have different local strata conditions (i.e., aquifer type). Further investigation of hydrogeology needs to be done to better understand the spatial heterogeneity of the aquifer system structure in the future. Furthermore, a positive trend for both land uplift and groundwater level rise was observed in the P3 well, which is located outside the S2 uplifting bowl.

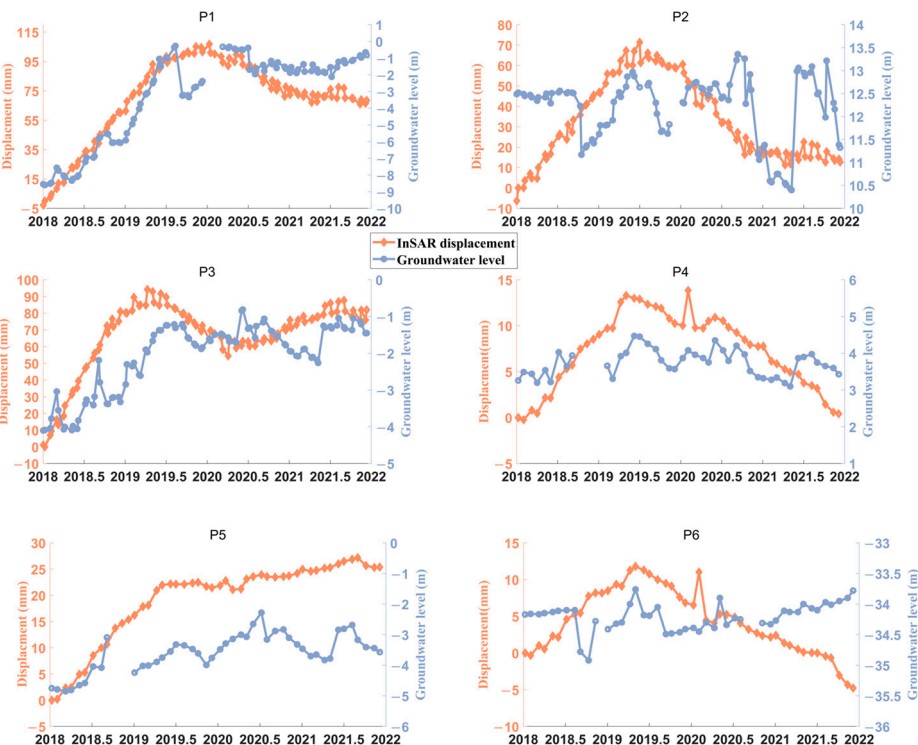

**Figure 9.** Ground deformation time series compared to groundwater level time series for nearby pixels (refer to Figure 5 for the specific locations of P1–P6).

In the slightly uplifting bowl of S3, wells P4, P5, and P6 exhibit groundwater level variability in accordance with the estimated displacement. Small-magnitude uplift was observed at P4, with a measured ground uplift of +0.6 mm, and an agreed slight upward trend in groundwater level change can be recognized (+0.42 m). The well P5, with higher changes in piezometric level, has experienced higher ground uplift. Thus, ground surface uplift and groundwater level rise generally correspond to each other. However, despite P6's proximity to P5, its time series exhibits differences from that of P5. More importantly, a minor increase in groundwater level was noted after July 2019, which was not replicated to some extent by the measured displacement, but instead showed a continuous downward trend. This could be attributed to the delayed response of the aquifer system to changes in pore pressure.

Insufficient data from observation wells located near the central area of subsidence bowls hinder the analysis of the relationship between them in these regions. To better understand the effect of the variation in groundwater level on the long-term subsidence phenomena, it is recommended to deploy more distributed wells in the subsiding zone. Additionally, conducting in situ geodetic measurements would help validate the results obtained from InSAR.

Overall, the fluctuations in groundwater level and time series of displacement at five wells (i.e., P1, P3, P4, P5, and P6) show that land uplift and subsidence are closely related to groundwater level changes. The observed uplift phenomena can be attributed to the poroelastic rebound of the aquifer system under the unloading of effective stress associated with a rise in the water table, which is referred to as poroelastic rebound [60,61]. As a result, the deformation mechanism can be explained based on hydrogeology, emphasizing the hydrogeological factors that control the ground displacement rate. However, at P2, there appears to be no correlation between ground deformation and groundwater level, which may be due to the heterogeneity of the spatial distribution and thickness of aquifers and aquifer units, the aquifer type, or the sparseness of groundwater observation wells. Additionally, all the displacement time series show almost linear variations and an obvious seasonal correlation (i.e., with the summer monsoon) is absent, indicating that ground

deformation is predominantly due to human causes rather than natural process [62–64]. Therefore, it can be proposed that government regulation and legal enhancement can effectively mitigate and arrest land subsidence. In the future, to better understand the effect of groundwater level variation on long-term subsidence phenomena, more distributed wells need to be deployed in the deformation zone, and more in situ geodetic measurements need to be conducted to validate the InSAR results.

### 5.3. Geological Factor Control Deformation Pattern

Surface geological formation, such as Quaternary deposition conditions, is an important inner geological background for ground deformation [65]. The Quaternary deposit is subject to significant settlement and rebound deformation under the effect of effective stress change due to fluctuations in groundwater level. As previously mentioned, the subsurface underlying the Plain can generally be divided into bedrock in mountainous and hilly environments and Quaternary sediments (Figure 1). Quaternary sediments are widely distributed in the Plain, covering an area of 1005 km$^2$. A strong spatial correlation between variations in land deformation and the distributions of Quaternary sediments can be observed through comparing the subsidence map during two monitoring periods (Figure 5) with the geological maps (Figure 1). Almost all of the deformation, including both uplift and subsidence, is distributed in lower-lying areas that are mainly filled by Quaternary deposits, particularly those with notable deformation rates. Conversely, the bedrock hill area shows a relatively stable and homogeneous pattern with very low or even zero deformation rates. Therefore, the distribution of deformation on soft soil suggests that the geological condition provides an environment for surface deformation.

On the other hand, the thickness of Quaternary sediments plays an important role in deciding the progression of land deformation [66]. To examine the spatial relationship between deposit thickness and deformation, a Quaternary sediment thickness map of the study area was constructed based on the depths obtained from boreholes drilled, using an ordinary Kriging interpolation method [67]. As seen in Figure 10a, an obvious correlation between them is observed. Subsidence rates decreased with increasing thickness, and the deeper deposits exhibited higher subsidence rates. Compression deformation is caused by excessive groundwater withdrawal in the Quaternary sediment layer. However, there is no correlation observed when the thickness exceeds 90 m, which could be the result of a combination of soft soil compaction and groundwater recovery. During the 2018 to 2021 period, the rising groundwater level increased pore water pressure, resulting in expansion of the sediments. Therefore, it is expected that higher uplift rates occur on thicker sediment, as evident in Figure 10b, indicating a close relationship between concentrated uplift and thicker soil layers. Soft soils thicker than 90 m are located in the central part of the Plain towards the northwest (Figure 2), and the land uplift also occurs in these places (Figure 5b). The uplift rate estimated for thicker deposits is relatively higher, and the maximum uplift coincides with the thickest. Therefore, changes in groundwater level are expected to be the main triggering factors in that area.

### 5.4. The Effect of Urban Construction

Urbanization has been often considered as an additional factor contributing to consolidation and subsidence processes [68]. The LJP, an industrial area of Guangdong province, has experienced rapid development in the textile and garment industry, which has become a perennial "locomotive" pulling the steady development of the Plain's economy. The rapid growth of the industry in terms of driving economic development has led to extensive construction of buildings, resulting in the imposition of significant loads on the ground and uneven settlement. The relationship between subsidence and urban construction, therefore, needs to be analyzed in detail.

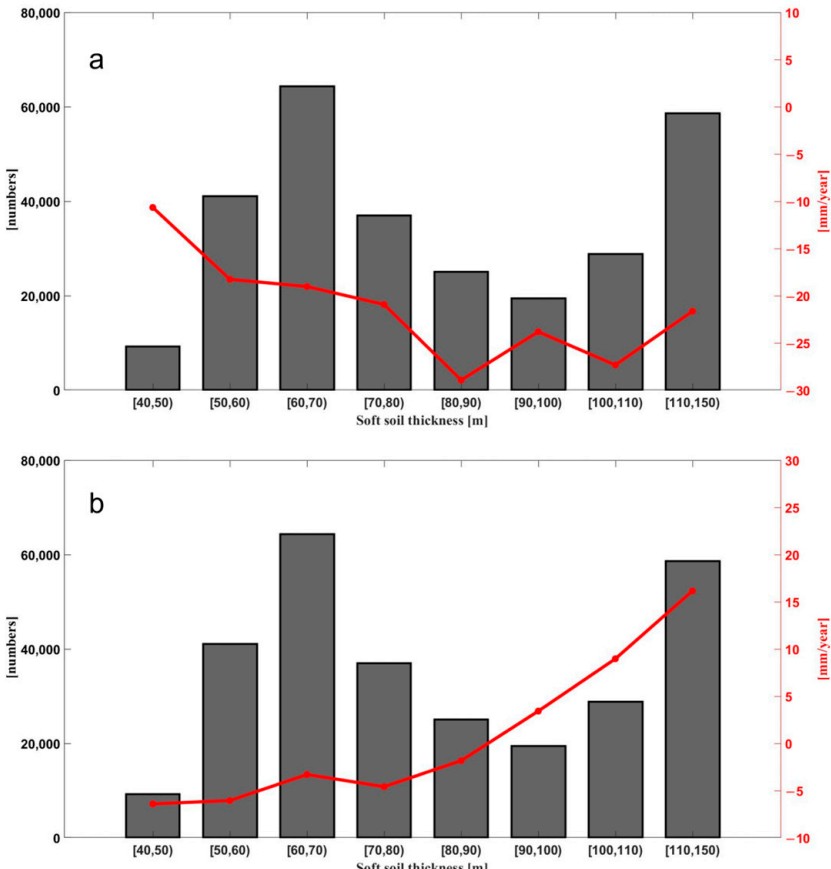

**Figure 10.** (**a**) Relationship between the deposit thickness with the deformation rates during 2015–2017 and(**b**) during 2018–2021. The average values of deformation velocity (red dots) are represented. Black bars represent the pixel number in the thickness layers.

To analyze the relationship between surface subsidence and urban construction, a visual interpretation of historical optical images was conducted to identify changes in buildings. These modified buildings were subsequently utilized as indicators of urban construction. Buildings constructed before 2015 were identified as unchanged areas, indicating no construction activities over the study period. In contrast, buildings con-structed between 2015 and 2021 were deemed as typical construction areas. In this study, three small construction areas (R1–R3 in Figure 5b) with significant building changes were selected for analysis. The distribution of land subsidence during 2018–2021 within three representative areas of building changes are shown in Figure 11a–c. Severe and uneven subsidence was observed predominantly within building construction areas, suggesting that urban construction may exacerbate and contribute to the development of land subsidence. Figure 11d–i present a comparison of historical images depicting building changes in regions R1–R3 over the five years. However, subsidence is not solely influenced by urban construction. In the case of region R2, subsidence was also observed in the unchanged area, suggesting that other factors may contribute to subsidence, and urban construction can contribute to the acceleration of the deformation. In the future, it is important to pay attention to newly constructed buildings as well as to the regular monitoring of existing buildings.

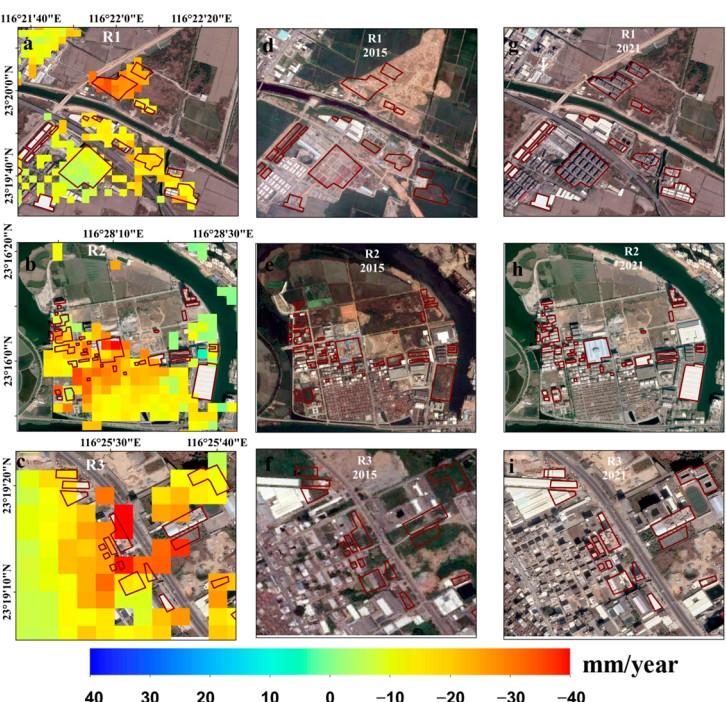

**Figure 11.** The deformation rate of building changes between 2015 and 2021 in three representative regions (**a**–**c**). The building change areas are denoted by red circles. (**d**–**i**) The comparison of optical images between 2015 and 2021 for region R1 to R3.

## 6. Conclusions

This study utilized Sentinel-1 SAR data of adjacent tracks to map surface deformation in the LJP during 2015–2021. The LOS velocity field was calculated over two observation intervals (2015–2017 and 2018–2021) to investigate the temporal behavior of land deformation phenomena. There was significant large-scale subsidence (more than −124 mm/year) from 2015 to 2017. However, the magnitude of subsidence decreased from 2018 to 2021, compared to the time span of 2015–2017. The subsidence rate decreased with a maximum subsidence of −45 mm/year, and three uplift zones were detected with a maximum uplift rate of up to +48.9 mm/year during 2018–2021. The Sentinel datasets from adjacent orbits acquired between 2018 and 2021 were cross-validated. The standard deviation of the difference between the two datasets is 2.4 mm/year. A variety of casual variables, including groundwater level change, thickness of Quaternary sediment, and urban construction, have been investigated. Comparisons between trends of deformation and groundwater level at six observation water wells within or near the uplift zone show some correlation between the change in groundwater level and deformation time series. The distribution of InSAR-derived deformation on Quaternary sediments suggests a spatial correlation between geological factors and deformation. The spatial thickness of Quaternary sediments was determined, and deformation was found to commence in areas where the sediment thickness is above 90 m and proportional to the thickness of the deposit. In addition, subsidence was found to be correlated with urban construction. Alteration in groundwater level is anticipated to be the primary influencing factor in land deformation. These findings provide further insights into understanding the cause of land deformation in the LJP and guide local authorities to take appropriate action. Ongoing monitoring of ground deformation will be essential for managing future surface deformation and safety control in the LJP and nearby cities.

**Author Contributions:** Conceptualization, Y.H. and A.H.-M.N.; Methodology, Y.H., A.H.-M.N. and H.W.; Validation, Y.H.; Formal analysis, Y.H. and J.K.; Investigation, Y.H., H.W. and J.K.; Writing—original draft, Y.H.; Writing—review & editing, A.H.-M.N., H.W. and J.K.; Supervision, A.H.-M.N. and H.W. All authors have read and agreed to the published version of the manuscript.

**Funding:** This research is funded by the Program for Guangdong Introducing Innovative and Entrepreneurial Teams (2019ZT08L213), National Natural Science Foundation of China (Grant no. 42274016/D0402), Natural Science Foundation of Guangdong Province (grant number 2021A1515011483).

**Data Availability Statement:** The Sentinel-1 data used in this study are downloaded from the European Space Agency (ESA) through the ASF Data Hub website https://vertex.daac.asf.alaska.edu (accessed on 20 March 2023). The DEM data used in the study is available at https://earthexplorer.usgs.gov/ (accessed on 10 March 2023). The Groundwater data used in the study is available at https://geocloud.cgs.gov.cn/ (accessed on 15 February 2023).

**Acknowledgments:** The authors would like to express gratitude to the European Space Agency for providing open access to the Sentinel-1 data used in this study.

**Conflicts of Interest:** The authors declare no conflicts interest.

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
