# Peer review of "Understanding the Spatiotemporal Characteristics of Land Subsidence and Rebound in the Lianjiang Plain Using Time-Series InSAR with Dual-Track Sentinel-1 Data"

_remotesensing, doi:10.3390/rs15133236_

Round 1

Reviewer 1 Report

"Authors have presented a study on “Land Subsidence at the Lianjiang Plain between 2015 and 2021 with dual track Sentinel-1 data” using time series interferometric synthetic aperture radar (TS-InSAR) technique and the correlation with the changes in groundwater level, geological factors, and urban construction are discussed herewith. Overall, the study is good but needs some improvement in the structuring of the manuscript. In order to enhance the scope of the manuscript, the following points could be considered.

11. The title of the manuscript needs to be reframed.

22. Introduction section could be improved by incorporating more literature part. Generally, it may include four paragraphs, a brief overview, literature, the research gap, and the objectives of the study. Currently, it is very concisely combined.

33.  Define the objectives as the major objectives of the study included (a)…; (b)…. In the last paragraphs of the Introduction section.

4 4. Methodology needs to be explained via a flow chart. It will help the reader to understand the concept easily.

5 5.Table 3 needs to be reframed with more literature related to SAR data usage. Also, you must name your study with some method. Spelling mistake in writing Sentinle-1 in Table 1.

66. Revise the paragraph under 5.2 heading. See the highlights and clearly mention your findings.

Detailed information about groundwater extraction at regional scale from 2015 to 2021 is publicly available (Error! Reference source not found.). Although it cannot accurately reveal the spatial-temporal variation of groundwater because of its coarse resolution, it does provide some valuable clues that the deformation pattern correlates with the groundwater extraction trend. Previous studies [29] have shown that the groundwater extraction data can roughly demonstrated that InSAR surface deformation induced by groundwater extraction. In addition to the regional groundwater data, groundwater level data of six aquifer wells (local scale) for the period 2018-2021 were also obtained for better spatial and temporal analysis of deformation induced by groundwater. As such, groundwater level changes at six groundwater observations wells nearby the uplift zones (Error! Reference source not found.(b)) were compared with the average ground deformation time series from all PS pixels in a radius of 150 m circle centered each observation well. The locations of groundwater wells stations in the vicinity of the uplift zones are shown in Error! Reference source not found. Considering that these observation wells are not located in proximity to zones of maximum uplift, thus, groundwater level trends from observation wells near the uplift zone were generally used to represent the overall groundwater level trends in and around the uplift zone.

77. Why this line is there in caption of figure 8 “The groundwater wells are marked by purple dots in Figure 4”?

88. Legend is not given in figure 5.

99. In 4.2 heading you have mentioned “overlap time and coverage of dataset II and dataset III enabled validation of the InSAR results by comparing the common part of deformation maps from both datasets”. Is this the only way to do validation? Can’t we compare the results with some another dataset “if the authors did not have access to the ground truths data over the site”.

110. Add some more references to strengthen your findings.

Round 2

Reviewer 1 Report

The authors have presented the revised version very well.

Table 2 is mistakenly mentioned as Table 1, just see that.